# Patient preferences for HIV service delivery models; a Discrete Choice Experiment in Kisumu, Kenya

Raphael Onyango Mando[1]*, Michelle Moghadassi[2], Eric Juma[1], Cirilus Ogollah[1], Laura Packel[3], Jayne Lewis Kulzer[2,3], Julie Kadima[1], Francesca Odhiambo[1], Ingrid Eshun-Wilson[4], Hae-Young Kim[5], Craig R. Cohen[2], Elizabeth A. Bukusi[1,2,3], Elvin Geng[4]

1 Research Care and Training Program, Center for Microbiology Research, Kenya Medical Research Institute, Nairobi, Kenya, 2 Department of Gynecology, Obstetrics, and Reproductive Sciences, University of California San Francisco, California, United States of America, 3 The University of California Berkeley, Berkeley, California, United States of America, 4 Division of Infectious Diseases, Department of Internal Medicine, Washington University in St. Louis, St. Louis, Missouri, United States of America, 5 School of Medicine, New York University, New York, New York, United States of America

* raphaelmando@icloud.com

**Data Availability Statement:** All data is available in the Dryad dataset repository at https://datadryad.org/ as follows: https://doi.org/10.5061/dryad.crjdfn37g.

## Abstract

Novel "differentiated service delivery" models for HIV treatment that reduce clinic visit frequency, minimize waiting time, and deliver treatment in the community promise retention improvement for HIV treatment in Sub-Saharan Africa. Quantitative assessments of differentiated service delivery (DSD) feature most preferred by patient populations do not widely exist but could inform selection and prioritization of different DSD models. We used a discrete choice experiment (DCE) to elicit patient preferences of HIV treatment services and how they differ across DSD models. We surveyed 18+year-olds, enrolled in HIV care for ≥6 months between February-March, 2019 at four facilities in Kisumu County, Kenya. DCE offered patients a series of comparisons between three treatment models, each varying across seven attributes: ART refill location, quantity of dispensed ART at each refill, medication pick-up hours, type of adherence support, clinical visit frequency, staff attitude, and professional cadre of person providing ART refills. We used hierarchical Bayesian model to estimate attribute importance and relative desirability of care characteristics, latent class analysis (LCA) for groups of preferences and mixed logit model for willingness to trade analysis. Of 242 patients, 128 (53.8%) were females and 150 (62.8%) lived in rural areas. Patients placed greatest importance on ART refill location [19.5% (95% CI 18.4, 10.6) and adherence support [19.5% (95% CI 18.17, 20.3)], followed by staff attitude [16.1% (95% CI 15.1, 17.2)]. In the mixed logit, patients preferred nice attitude of staff (coefficient = 1.60), refill ART health center (Coeff = 1.58) and individual adherence support (Coeff = 1.54), 3 or 6 months for ART refill (Coeff = 0.95 and 0.80, respectively) and pharmacists (instead of lay health workers) providing ART refill (Coeff = 0.64). No differences were observed by gender or urbanicity. LCA revealed two distinct groups (59.5% vs. 40.5%). Participants preferred 3 to 6-month refill interval or clinic visit spacing, which DSD offers stable patients. While DSD has encouraged community ART group options, our results suggest strong preferences for

**Funding:** President's Emergency Plan for AIDS Relief (PEPFAR) through the Centers for Disease Control and Prevention (CDC) under the terms of U2GGH001947 to CRC and EAB. The funders had no role in study design, data collection and analysis, decision to publish, or preparation of the manuscript.

**Competing interests:** The authors have declared that no competing interests exist.

**Abbreviations:** AIC, Akaike's Information Criterion; ART, Anti-Retroviral Therapy; BIC, Best Information Criterion; CAIC, Criterion Akaike's Information Criterion; CDC, Centre for Disease Control; CI, Confidence Interval; DCE, Discrete Choice Experiment; DSD, Differentiated Service Delivery; DSM, Differentiated Service Models; HIV, Highly Immunodeficiency Virus; IRR, Incident Rate Ratios; KEMRI, Kenya Medical Research Institute; KENPHIA, Kenya Population-based HIV Impact Assessment; LCA, Latent Class Analysis; LMIC, Low- and Middle-Income Countries; PLHIV, Persons Living with HIV; SERU, Scientific Ethics Regulatory Unit; UCSF, University of California in San Francisco.

ART refills from health-centers or pharmacists over lay-caregivers or community members. These preferences held across gender&urban/rural subpopulations.

## Background

Kenya has over 1.3 million adults living with HIV and among those that know their status, nearly all (96%) are on lifesaving antiretroviral therapy (ART) [1] This is a steep increase from a few years ago when only 68% of the eligible people were on ART [2]. In 2015, the World Health Organization (WHO) recommended 'test and treat'—immediately initiating ART for all who test HIV positive, and Kenya followed suit, adopting the WHO guidelines in 2016 [3]. Implementing test and treat in high HIV burden regions of Kenya, such as Kisumu County where HIV prevalence is 17.5%, compared to the nationwide prevalence of 4.9% [1], has led to overstretched health systems. Clinics are now faced with pressure to provide quality services with steep increases in patient volume, overseeing ART management for both new and returning patients, while juggling inadequate human resources. In this context, it is essential to optimize HIV care services so that health systems can successfully manage both new patients initiating ART and stable patients in care.

Differentiated service delivery models (DSMs), which vary the timing, frequency, location, and providers in delivering care, have been singled out as a strategic solution to both improve the care cascade and enhance efficiency of HIV care. DSMs attempt to reduce system inefficiencies where clinics and staff are often overburdened and facilities overcrowded by reducing visits, using lay health workers, and encouraging patient-based groups and community-based treatment [4, 5].

DSMs have been successfully implemented and shown to improve retention of stable patients in care including children, adolescents, and key populations [4–6]. DSMs are designed to tailor care to meet the unique needs of patients regardless of their age, socioeconomic status, background, viral load levels, and stability on treatment (e.g., poor adherence, not virally suppressed) [7].

Kenya rolled out the Differentiated Operational Guide and DSM services in 2017 [7]. With the implementation of differentiated models for HIV care in Kenya, we sought to gain a deeper understanding of which aspects of clinical care were most important to patients. Garnering patient preferences provides essential input in the context of differentiated service delivery planning, with the goal of designing, implementing, and refining care models that work well for both patients and staff, and lead to high patient retention and good health outcomes [6]. The literature is limited regarding patients' preferences as they relate to differentiated care models. To bridge this gap, we conducted a Discrete Choice Experiment (DCE) to explore which attributes of HIV care and treatment are most important to patients receiving their care in Kisumu, Kenya.

This study expands on our previous DCE analyses in Kenya [8]; using the same methodology, we increased the sample size and included participants from four health facilities within Kisumu, two high volume facilities and two low volume facilities.

## Methods

### Ethics statement

This research obtained Institutional Regulatory Board (IRB) approval from the KEMRI Scientific Ethics Regulatory Unit (SERU), #1/2009 and the UCSF IRB, #11–05348. It was also

reviewed in accordance with the US Centers for Disease Control and Prevention (CDC) human research protection procedures and obtained non-research determination approval (or determined to be research but CDC did not interact with human subjects or have access to identifiable data for research purposes. DCE participants provided verbal informed consent.

## Study design

We conducted a cross-sectional DCE over the course of four weeks, between February and March 2019, to understand HIV care preferences of adults living with HIV in Kenya. The DCE elicits individual preferences by offering respondents a set of choices based on pre-specified attributes and levels corresponding to those attributes and provides an opportunity to determine which combinations of attributes and levels respondents rank as most important [9].

## Study population and sampling

The study was conducted in a high prevalence region of Kenya. Kisumu County, along the shores of Lake Victoria in western Kenya, has the second highest HIV prevalence in the country, over three times the national prevalence. The county is home to the third largest city in Kenya, Kisumu City, yet is largely rural beyond the city borders.

The study took place at four health clinics in Kisumu County supported by Family AIDS Care and Education Services (FACES). FACES is a collaboration between the University of California, San Francisco (UCSF), the Kenya Medical Research Institute (KEMRI), and the Kisumu County Ministry of Health (MOH).

The four health facilities were purposively selected based on the population of people living with HIV (PLHIV) and their geographical distribution across urban and rural facilities supported by the FACES program. The four facilities varied in size, ranging from serving more than 1,000 to under 500 PLHIV in care. One site was an urban hospital, and the other three sites were located in more rural areas.

Inclusion criteria for PLHIV were enrolled in HIV care at one of the four health facilities, age 18 years and older, having been on ART for six months or longer, willing to provide informed consent, and able to communicate in English, Kiswahili, or the Dholuo language. The exclusion criteria for the for PLHIV were not enrolled in HIV care one of the four health facilities, age below 18 years, having been on ART for less than six months, not willing to provide informed consent, and not able to communicate in English, Kiswahili, or the Dholuo language.

Potential participants were systematically selected for inclusion in the study at the pharmacy, immediately after dispensing of their medication. The interviewer selected every kth individual who exited the pharmacy and introduced themselves and the study.

## Measurements

Both sociodemographic and DCE variables were captured in this study. The questionnaire was anonymous; however, key sociodemographic information was obtained to characterize participants. These included age, gender, residence, income, and level of education.

Our investigation is an extension of initial study on Preferences of People Living with HIV for Differentiated Care Models in Kenya. Methods for selection of attributes are detailed elsewhere [8], but briefly, the team identified the attributes based on an extensive literature review of differentiated care models in place in sub-Saharan Africa. The DSD Models available in are (i). FastTrack ART delivery: This is a facility-based system for ART (and other medication) refills whereby the pharmacist prepares the medications the day prior for client drug pick-up

and the client proceeds directly to the pharmacy dispensing window, bypassing all other health care services, and reducing the overall time needed to acquire the refill. (ii). Facility Based ART Groups (FB-AG): This model uses a support-group structure to provide ART refills to clients. Each client in the FB-AG is required to come to the facility every 6 months for a clinical review appointment, with ART refills distributed through the FB-AG every 3 months between these facility appointments. (iii). Community-based ART Groups (CAGs): This model uses a support-group structure to provide ART refills to clients in the community. CAGs may be led by either a health care worker (HCW-led CAG), such as nurse or clinical officer, or a peer living with HIV, (peer-led CAG) [4, 5]. This was followed by qualitative interviews with the FACES differentiated care team and clinicians to further tailor the attributes to the specific context. Details about the attributes and levels included in the DCE are shown in Table 1.

The selected attributes reflect the elements of care that can be adjusted with various models of differentiated service delivery (Fig 1). For instance, the **number of required clinical visits** is an important care component that, when reduced, has been associated with a nearly two-fold odds of retention in care [10].

**Reduced frequency of ARVs pick-ups** has also shown a trend towards better retention and lowering the burden of care on both the patient and the facility resources [10, 11]. Multi-month scripting of patients [12] leading to fewer required visits to the health facility are another component of the care model that can be varied in differentiated models.

**Location of ART refill pickup**, such as community pharmacies or pharmacy-only refill programs are additional options that have been shown to be preferred by ART patients and lead to improved retention outcomes with little loss to follow-up. Pharmacy-only refills have

**Table 1. Attributes and levels included in the DCE.**

| Attribute | Levels |
|---|---|
| Location of ART refills | Health Center |
| | Community meeting point |
| | Home |
| Frequency of receiving ART refills | Every month |
| | Every 3 months |
| | Every 6 months |
| Adherence support provided | No support |
| | Individual support |
| | Group support |
| Refill pick-up/delivery times | Weekday during facility hours |
| | Weekday, early morning, or evening |
| | Weekend |
| Attitude of facility staff | Rude |
| | Nice |
| Frequency of clinical visits | Every month |
| | Every 3 months |
| | Every 6 months |
| | Every 12 months |
| Person providing ART refills | Nurse |
| | Lay health worker |
| | Pharmacist |
| | Person living with HIV (community peer) |

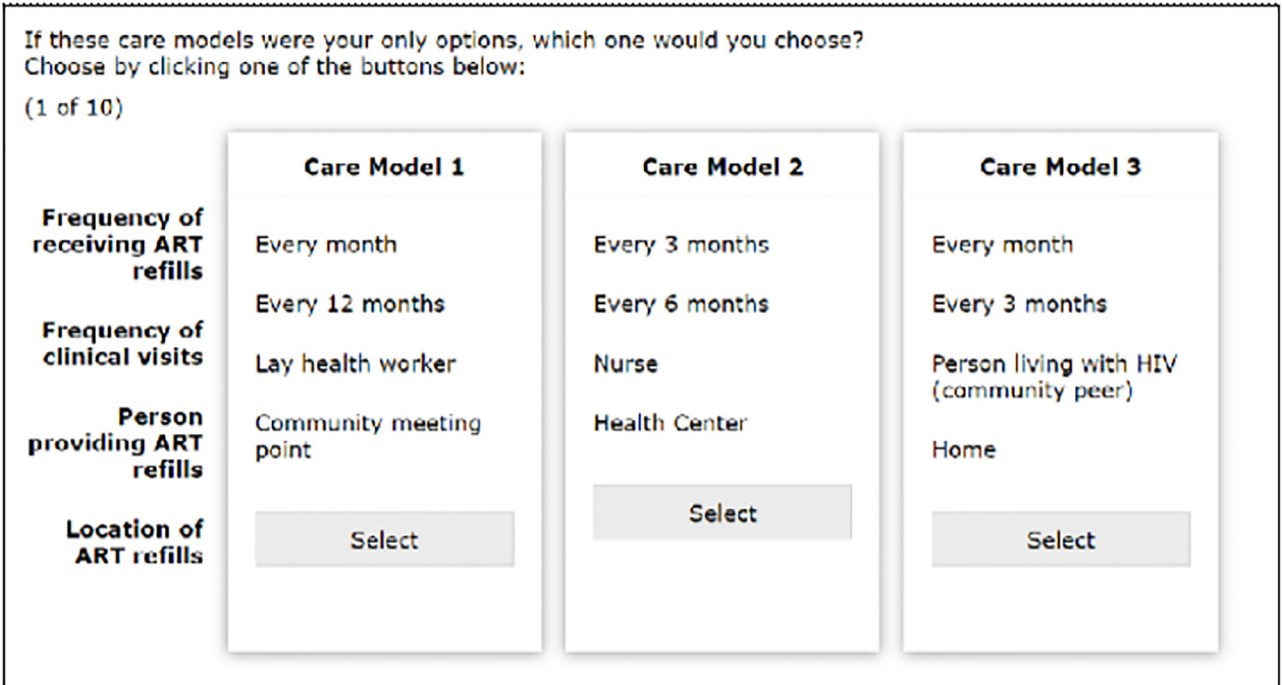

**Fig 1. Sample care models in the choices.**

demonstrated effectiveness in improving HIV outcomes for stable and adhering patients in high volume facilities [13–15].

**The qualifications of the person providing ART refills** is another attribute that differentiated care models can vary; Community ART Groups (CAGs) in which ART is provided by designated peers in the community is one such example [12]. Other examples of people qualified to provide ART refills are clinicians, pharmacists and self-collect e.g., through vending machines.

**Adherence support** is another important element of HIV care that can be varied in differentiated care models. Other work has indicated that HIV care recipients feel strongly that adherence support should always be provided to all patients [16], but less is known about preferences for implementation models for adherence support (e.g., individual or group support).

Finally, although not a component of care that is varied as part of differentiated models, the **attitude of clinical staff and the level of patient satisfaction** with how clinical staff treats them is an important aspect of care to consider in the context of how much weight or importance patients ascribe to the HIV care experience. Patients who are unsatisfied with staff attitude often unlikely to be satisfied with overall care quality [17], thus, staff attitude also needs to be taken into consideration when ranking importance of clinical care attributes to patients.

Each respondent received a randomly generated set of ten choice tasks (out of a total of 120 possible choice sets. Each questionnaire then consisted of ten choice tasks where each choice task was a set of three hypothetical care models, each with different levels of attributes. For each choice task, participants were asked to choose the care model they most preferred. Pre-randomized options for the choice tasks were generated from the Lighthouse Studio software using a Balanced Overlap technique, and the questionnaires were administered to participants using Android tablets in their preferred language (English, Kiswahili, or Luo).

## Analysis

### Attribute importance and part worth utilities

Analysis was conducted using Sawtooth Software 9.7.1 and STATA 16.0. The Hierarchical Bayesian model in Sawtooth was used to determine the attribute importance and respective utility values as selected by respondents.

Average importance of attributes was calculated using a choice based conjoint analysis in Sawtooth, reporting attribute importance as a percentage. Attribute importance calculation was used to determine how much the respondents preferred the DSD attributes that they were asked to choose from.

Sociodemographic characteristics were computed using counts and proportions in Stata.

### Willingness to trade analysis

We used the mixed logit model to generate preference weights across the attributes which were then used to perform the willingness to trade analysis [18]. A positive stated preference value of above 1 indicate a strong preference, below 1 but above 0.5 indicate a mild preference, while below 0.5 indicate a low preference.

All attribute levels were fitted as random to allow for heterogeneity in patient responses. The attribute levels included in the in the willingness to trade analysis were drawn from positive utility values from sawtooth software. These included (i) Location of ART Refill—Health Centre; (ii) Adherence support provision for Individuals; (iii) Adherence Support Provision—Group; and (iv) Attitude of Facility Staff—Nice. We fitted the final model using the mix logit model, keeping the attribute/level combinations as random effects. The McFadden Psuedo-R2 $(1 - [e(ll)/e(ll\_0)])$ was calculated to determine the goodness of fit of mixed logit model for the data. We performed the non-traditional willingness to trade analysis -a post estimation to further validate the initial willingness to trade analysis using linear combinations of estimators. From the attribute designated as willing to trade, we subtracted the coefficients of significant attributes, reporting for relative-rate ratio. A positive and significant coefficient of the willingness to trade indicate that the respondents are willing to trade.

### Latent class analysis

We also conducted a latent class analysis, using the latent class logit model, to determine the most appropriate set of latent classes—the packages of DSD services that were preferred by the respondents, specifically classifying the respondents based on the attribute/level combinations that they preferred- and their utility values.

### Inclusivity in global research

Additional information regarding the ethical, cultural, and scientific considerations specific to inclusivity in global research is included in S2 Text.

## Results

### Patient characteristics

In Table 2, between February 2019 and March 2019, we enrolled 239 patients enrolled in HIV care in four health facilities across rural and urban areas of Kisumu County in Kenya. Of the 239 PLHIV who completed the DCE, two were excluded because they had been in care for fewer than six months. Most of the respondents lived in urban areas (n = 89, 62.9%), and the median age among the respondents was 38, Interquartile Range (31–44). More than half of the

**Table 2. Sociodemographic characteristics of respondents.**

| Age | N | % | 95% CI | |
|---|---|---|---|---|
| Age (years; median and interquartile range) | 38 (31–44) | | | |
| **Income** | | | | |
| Below 5000 | 129 | 54.4 | 48.0 | 60.7 |
| 5000 and above | 109 | 45.8 | 39.5 | 52.2 |
| **Gender** | | | | |
| Male | 110 | 46.0 | 39.7 | 52.4 |
| Female | 128 | 54.0 | 47.6 | 60.3 |
| **Education** | | | | |
| primary education and below | 137 | 57.8 | 51.4 | 64.0 |
| Secondary education and above | 101 | 42.4 | 36.3 | 48.8 |
| **Frequency of Visits** | | | | |
| About every 6 months | 28 | 11.8 | 8.3 | 16.6 |
| About every 3 months | 142 | 59.9 | 53.5 | 66.0 |
| More often than every 3 months | 68 | 28.3 | 22.9 | 34.4 |
| **Travel Time** | | | | |
| 30 and below | 128 | 53.6 | 47.2 | 59.9 |
| 30–60 mins | 83 | 35.0 | 29.2 | 41.3 |
| above 60 | 27 | 11.4 | 7.9 | 16.1 |
| **Urbanicity** | | | | |
| Urban | 89 | 62.9 | 56.5 | 68.8 |
| Rural | 150 | 37.1 | 31.2 | 43.5 |

total respondents earned below the equivalent of $50 USD per month. Most respondents were receiving care every three months (142, 58.6%), most of the respondents reported having to travel for less than 30 minutes to get to the health facility (128, 52.9%), and the majority had a primary education or less (137, 56.6%).

## Importance of attributes to respondents

The conjoint analysis showed that (19.5%) of the respondents selected *location of ART refills* as the most important attribute and an equivalent proportion selected *provision of adherence support* as the most important attribute (Table 3). This was followed by *attitude of the facility staff* (16.14%) and *frequency of clinical visits* (14.76%), *frequency of receiving ART refills* (12.61%) and *the person providing the ART refills* respectively (11.21%). *Refill pick up/delivery times* was the least important attribute to the respondents (6.23%).

**Table 3. Importance of attributes.**

| Overall average Importance of Attributes | | | | |
|---|---|---|---|---|
| Attribute | Importance (%) | SD | Lower 95% CI | Upper 95% CI |
| Location of ART refills | 19.5 | 8.7 | 18.4 | 20.6 |
| Adherence support provided | 19.5 | 6.4 | 18.7 | 20.3 |
| Attitude of facility staff | 16.1 | 8.0 | 15.1 | 17.2 |
| Frequency of clinical visits | 14.8 | 5.0 | 14.1 | 15.4 |
| Frequency of receiving ART refills | 12.6 | 5.3 | 11.9 | 13.3 |
| Person providing ART refills | 11.2 | 3.7 | 10.8 | 11.7 |
| Refill pick-up/delivery times | 6.2 | 3.6 | 5.8 | 6.7 |

## Stated preferences for differentiated service delivery

Results from the mixed logit model on the stated preferences (Table 4) show that respondents held a strong preference for a nice attitude from a provider as opposed to a provider with a bad attitude (preference weight: β = 1.61; 95% CI:1.36 to 1.86). Patients preferred to pick up their ART refills at the health center over having them brought to their home or delivered by community ART Groups (preference weight: β = 1.58; 95% CI: 1.27 to 1.89). Patients also had a strong preference for individual adherence support compared to group adherence support or even no adherence support (preference weight: β = 1.55; 95% CI: 1.29 to 1.80). ART refill every three months was slightly preferred compared to 6 months (preference weight: β = 0.95; 95% CI: 0.73 to 1.18). Patients preferred clinical visits every 3 months compared to visits every 6 months (preference weight: β = 0.67; 95% CI: 0.45 to 0.89). Patients preferred the pharmacist as the person providing the refill over a nurse or the lay health worker (preference weight: β = 0.64; 95% CI: 0.42 to 0.86). Finally, patients ranked refill delivery times as least important, but preferred refill delivery on weekdays in the early morning and evening, (preference weight: β = 0.14; 95% CI: -0.06 to 0.35).

**Table 4. Mixed Logit Model—(a) Preferences and (b) Heterogeneity of Preferences for Differentiated Service Delivery Attributes.**

| Attributes | β | [95% Conf. Interval] | | SE | p-value |
|---|---|---|---|---|---|
| (a) Preferences | | | | | |
| Location of ART Refill-Health Centre* | 1.578 | 1.269 | 1.886 | 0.157 | <0.001 |
| Frequency of ART Refill-3months* | 0.953 | 0.730 | 1.176 | 0.114 | <0.001 |
| Frequency of ART Refill-6months | 0.816 | 0.571 | 1.061 | 0.125 | <0.001 |
| Adherence Support-Individual* | 1.545 | 1.287 | 1.803 | 0.132 | <0.001 |
| Adherence Support-Group | 1.411 | 1.136 | 1.686 | 0.140 | <0.001 |
| Attitude of provider-Nice* | 1.612 | 1.360 | 1.864 | 0.129 | <0.001 |
| Frequency of Clinical Visists-3 months* | 0.673 | 0.453 | 0.893 | 0.112 | <0.001 |
| Frequency of Clinical Visists-6months | 0.485 | 0.252 | 0.718 | 0.119 | <0.001 |
| Person Providing Refills-Nurse | 0.409 | 0.168 | 0.650 | 0.123 | 0.001 |
| Person Providing Refills-Pharmacist* | 0.642 | 0.429 | 0.856 | 0.109 | <0.001 |
| Refill Delivery times -Weekday during facility hours | 0.043 | -0.174 | 0.261 | 0.111 | 0.697 |
| Refill Delivery times -Weekday early Morning and evenings* | 0.145 | -0.060 | 0.350 | 0.105 | 0.167 |
| Location of ART Refill-Community Meeting Point | -0.101084 | -0.3435435 | 0.141376 | 0.123706 | 0.414 |
| **Attributes** | **β** | **[95% Conf. Interval]** | | **SE** | **p-value** |
| (b) Heterogeneity across preferences | | | | | |
| Location of ART Refill-Health center | 1.427 | 1.121 | 1.734 | 0.156 | <0.001 |
| Frequency of ART Refill-3months | 0.058 | -0.160 | 0.277 | 0.112 | 0.601 |
| Frequency of ART Refill-6months | 0.590 | 0.297 | 0.884 | 0.150 | <0.001 |
| Adherence Support-Individual | 0.621 | 0.327 | 0.915 | 0.150 | <0.001 |
| Adherence Support-Group | 0.748 | 0.431 | 1.065 | 0.162 | <0.001 |
| Attitude of provider-Nice | 1.231 | 0.884 | 1.578 | 0.177 | <0.001 |
| Frequency of Clinical Visists-3 months | 0.519 | 0.107 | 0.930 | 0.210 | 0.013 |
| Frequency of Clinical Visists-6months | 0.566 | 0.011 | 1.121 | 0.283 | 0.046 |
| Person Providing Refills-Nurse | 0.711 | 0.319 | 1.102 | 0.200 | <0.001 |
| Person Providing Refills-Pharmacist | 0.359 | -0.325 | 1.042 | 0.349 | 0.304 |
| Refill Delivery-Weekday during facility hours | 0.001 | -0.409 | 0.412 | 0.209 | 0.996 |
| Refill Delivery-Weekday early Mornings and evenings | 0.352 | 0.059 | 0.646 | 0.150 | 0.019 |
| Location of ART Refill-Community Meeting Point | -0.366 | -0.783 | 0.050 | 0.213 | 0.085 |
| Model Specifications: | Log likelihood = -2016.3418 | Wald chi2(13) = 397.45 | Prob > chi2 < 0.0001 | | |

## Willingness to trade

We assessed the respondent's willingness to trade certain attributes for the others using the traditional willingness to trade analysis in S1 to S8 Tables. A positive preference value of above 1 indicate a strong preference, below 1 but above 0.5 indicate a mild preference, while below 0.5 indicate a low preference. Respondents were unwilling to trade health center as a preferred location for ART refill (Coeff = -8.167, p-value<0.001, 95% CI[-9.117,-7.216]), individual adherence support (Coeff = -6.752, p<0.001, 95% CI [-7.545,-5.960]) and group adherence support (Coeff = -7.1598, p<0.001, 95%CI [7.956, -6.364]) or even a nice attitude of staff for the attributes (Coeff = -6.58, p<0.001, 95% CI [-7.39,-5.77]).

## Latent class analysis

We achieved clear convergence for an optimal/modest number of two (2) classes based on our sample size. We included sociodemographic characteristics to predict class membership, and a two-class model with five sociodemographic covariates was chosen.

The class probabilities indicated that 40.5% of respondents were assigned to class 1 (the smaller class) and 59.5% were assigned to class 2 (the larger class). Table 5 presents the class specific **β**-values weights of the seven attributes in the two classes.

The preference weights for the attribute levels between Class 1 and Class 2 were significantly different as follows; Preference of Location of ART Refill -health Centre ($\Delta\beta$ = 1.38), Frequency of ART Refill -6 monthly ($\Delta\beta$ = 0.805), Frequency of ART Refill -3 monthly ($\Delta\beta$ = 0.563), adherence support provided individual ($\Delta\beta$ = 0.187), Adherence Support provided -Group (-0.563, Frequency of Clinical Visits -3 months ($\Delta\beta$ = 0.083), Person Providing Refill -Pharmacist ($\Delta\beta$ = -0.026), Location of Art Refill -Home ($\Delta\beta$ = -.841), and Preference of a Nice attitude of a care provider ($\Delta\beta$ = -1.55).

In Table 6, we found gender to be the only significant predictor of larger class membership. Female gender was 2.4 times likely to predict large class membership compared to being male (Relative Risk Ratio [RRR] = 2.415, 95% CI [1.377 to 4.234]). Age, urbanity, travel time and education were not significantly associated with class membership. The model had a strong

**Table 5. Latent class analysis model for two classes with confidence intervals.**

| Choice | Class 1 | | Class 2 | |
|---|---|---|---|---|
| | β | p-value | β | p-value |
| Location of ART Refill -Health Centre | 2.164 | <0.001 | 0.784 | <0.001 |
| Location of ART Refill -Home | -0.506 | 0.029 | 0.335 | 0.016 |
| Frequency of ART Refill—3 monthly | 1.214 | <0.001 | 0.675 | <0.001 |
| Frequency of ART Refill—6 monthly | 1.272 | <0.001 | 0.467 | 0.001 |
| Adherence Support provided—Individual | 1.506 | <0.001 | 1.319 | <0.001 |
| Adherence Support provided—Group | 0.893 | <0.001 | 1.456 | <0.001 |
| Attitude of Care Provider -Nice | 0.427 | 0.033 | 1.977 | <0.001 |
| Frequency of Clinical Visits—3 months | 0.700 | 0.001 | 0.617 | <0.001 |
| Frequency of Clinical Visits—6 months | 0.715 | 0.002 | 0.273 | 0.079 |
| Frequency of Clinical Visits—12 months | 0.318 | 0.141 | -0.030 | 0.851 |
| Person Providing Refill Nurse | 0.294 | 0.120 | 0.437 | 0.001 |
| Person Providing Refill -Pharmacist | 0.500 | 0.003 | 0.526 | <0.001 |
| Refill Pick Up/Delivery times -Weekday during facility hours. | 0.246 | 0.216 | -0.079 | 0.547 |
| Refill Pick Up/Delivery times -Weekday early mornings and evenings | 0.621 | 0.001 | -0.116 | 0.346 |
| Model Specifications | AIC 4110.384 | CAIC 4239.96 | BIC 4210.96. | '-2Log-likelihood -2026.19 |

**Table 6. Latent class membership predictors reporting relative-risk ratios.**

| class | RRR | p-value | [95% Conf. Interval] | |
|---|---|---|---|---|
| 1 | (Base outcome) | | | |
| 2 | | | | |
| Age groups | | | | |
| 18-24yrs | 1 | | | |
| 25-29yrs | 1.826 | 0.339 | 0.531 | 6.277 |
| 30-34yrs | 1.761 | 0.341 | 0.549 | 5.651 |
| 35–39 yrs. | 1.613 | 0.407 | 0.521 | 4.994 |
| 40 = 44yrs | 1.299 | 0.656 | 0.412 | 4.095 |
| 45-49yrs | 1.751 | 0.393 | 0.484 | 6.331 |
| 50 plus | 1.805 | 0.331 | 0.548 | 5.945 |
| Urbanity | | | | |
| urban | 1.000 | | | |
| rural | 1.402 | 0.266 | 0.773 | 2.545 |
| Travel Time | | | | |
| 0–30 mins | 1.000 | | | |
| 30–60 mins | 1.210 | 0.536 | 0.661 | 2.215 |
| above 60 | 1.328 | 0.548 | 0.527 | 3.346 |
| Gender | | | | |
| Male | 1.000 | | | |
| Female | 2.415 | 0.002 | 1.377 | 4.234 |
| Education | | | | |
| primary and below | 1.000 | | | |
| Secondary education | 1.531 | 0.170 | 0.833 | 2.812 |
| tertiary education | 1.386 | 0.578 | 0.440 | 4.366 |

predictive ability, predicting class membership with up to 90% certainty. See S9 Table for more details.

## Discussion

In this study of PLHIV patient preferences for DSD components, respondents had the strongest preference for the health center as the preferred location for ART refill, nice provider attitude, and adherence support provision at the individual level and group level. The most important attributes were refill in the health facility, adherence support, and nice attitude and the primary attributes participants did not want to trade for any other attributes included *health centers as the preferred location for ART refills*, *individual or group adherence support*, and a *nice staff attitude*. The preference for health facility for refills is substantiated by other studies [19–21]. This may be due to more normalization of routine care and familiarity of care provision. It may also suggest perceived or anticipated stigma in the community, whereas protection of privacy in the health center may feel more secure. This is substantiated by a recent study conducted in Ghana that found that fear of stigma and discrimination was very strong and the main barrier to community-based models [21].

The importance of a friendly provider attitude is notable as a priority for patients. Patients in our study preferred a nice attitude of staff. This preference comes out strongly in other DCEs [9, 22]. A study in Zimbabwe also showed that patients valued care providers who gave them respect and understanding over all other care attributes [23, 24]. Moreover, in resource limited settings, poor staff attitudes are an important barrier to ART initiation and adherence

[24]. It is not just important to pay attention to patient preferences but also that our findings were consistent with other studies that patients strongly prefer health facilities over CAG models (for whatever reason) and programs should consider shifting DSDs toward that.

The strong preference for adherence support observed in our study was expected as other studies have shown that adherence support improves provider and patient communication, adherence, and viral suppression [25]. Similar to findings by Sagar [8], our participants preferred adherence support in general, but preferred at the individual level over group level. This may be attributed to individual consultation that comes with individual adherence support [22]. Interestingly, two of the three top preferences represent aspects of HIV care outside of the DSM package. Catering to patient preferences as part of DSM may help patients stay in care, for example strengthening warm, welcoming staff, and bolstering adherence support [21].

Milder preferences were in line with DSM packages, for instance, a slight preference for refills at three over six months and for clinical visits at three rather than six months, as opposed to every month. These findings are in line with two other studies, one in Zambia and the other in urban Zimbabwe where patients preferred three months or longer pick-up schedules to monthly pickups [22, 26]. Qualitative investigations have also shown that shorter refill times, (i.e., 30 days) causes anxiety while longer refill times reduce those anxieties [27]. There was a mild preference for refill by a pharmacist rather than a nurse or lay health care worker, and ART refill at a community meeting point was least preferred. Patients in pharmacy-managed clinics in China were shown to have better medication adherence ≥80% compared to standard adherence support group or a control group [28, 29]. Patients may prefer pharmacy dispensation due to the individual level consultation associated with it [22].

Of note, *refill pick up/delivery times* were the least important attribute to the respondents. While there was a slight preference for early weekday mornings or weekday evenings, flexibility here is helpful to note. Offering a wider window for access can ensure patients with varied schedules who are willing to retrieve their medication during less traditional hours can do so and avoid pauses in treatment. Our study confirms the findings of a study in Zimbabwe that measured clinic opening times and found that they were not significant drivers of patient preferences [22].

The main limitation of our investigations is on the generalizability of the findings beyond the care settings in Kenya. It could be also that few of our respondents were in CAGs and so their views may be underrepresented because fast-track is more widely adopted and therefore fewer participants had experience with CAGs. The main strength of this study is in the DCE design which allows for simulating reality based on hypothetical situations. Most of the options offered to the respondents as choices were also based on a qualitative study (key informant interviews and focused group discussions) that were conducted at one of the largest cares and treatment facilities in Western Kenya -Lumumba Sub County Hospital [8], backed by extensive literature review. Our sample size was also representative for the region since it included respondents from both large and small care facilities in rural and urban areas.

## Conclusion

These findings help may guide future design of DSD models with the dual goals of reducing clinic burden and increasing patient retention. While all attributes can be taken into consideration in the design of programs, the order and strength of preferences is important and should be integrated into program design, including attributes within the DCE model (health facility for medication pick up) and outside of the DCE model (nice provider attitude and adherence support). This is especially salient given that lack of key attribute preferences may deter an

individual from remaining in care. To strengthen uninterrupted treatment with the goal of viral load suppression, reduced coinfections and comorbidities, and decreased population level incidence of HIV and deaths due to HIV-related causes, these preferences are tantamount and should serve as the foundation for the design future DSD models for HIV treatment.

## Supporting information

**S1 Table. Willingness to trade location (health centre for) other attributes.**
(XLSX)

**S2 Table. Willingness to trade location of art refill health centre.**
(XLSX)

**S3 Table. Willingness to trade individual adherence support provided for other attributes.**
(XLSX)

**S4 Table. Willingness to trade adherence support for individual.**
(XLSX)

**S5 Table. Willingness to trade group adherence support provided for other attributes.**
(XLSX)

**S6 Table. Willingness to trade group adherence support.**
(XLSX)

**S7 Table. Willingness to trade good attitude of staff for other attributes.**
(XLSX)

**S8 Table. Willingness to trade a nice attitude.**
(XLSX)

**S9 Table. How good is the model.**
(XLSX)

**S1 Checklist. STROBE statement—Checklist of items that should be included in reports of cross-sectional studies.**
(DOCX)

**S1 Text. Attribute importance and part worth utilities.**
(DOCX)

**S2 Text. Inclusivity in global research.**
(DOCX)

## Acknowledgments

The authors appreciate the participants for their valuable time, facility staff, KEMRI Director, and Family AIDS Care and Education Services (FACES) for their support of this Discrete Choice Experiment.

## Author Contributions

**Conceptualization:** Raphael Onyango Mando, Michelle Moghadassi, Jayne Lewis Kulzer, Julie Kadima, Francesca Odhiambo, Craig R. Cohen, Elizabeth A. Bukusi, Elvin Geng.

**Data curation:** Raphael Onyango Mando, Eric Juma, Ingrid Eshun-Wilson.

**Formal analysis:** Raphael Onyango Mando, Ingrid Eshun-Wilson.

**Funding acquisition:** Jayne Lewis Kulzer, Craig R. Cohen, Elizabeth A. Bukusi, Elvin Geng.

**Investigation:** Michelle Moghadassi, Eric Juma, Cirilus Ogollah, Julie Kadima, Francesca Odhiambo, Elvin Geng.

**Methodology:** Michelle Moghadassi, Cirilus Ogollah, Jayne Lewis Kulzer, Julie Kadima, Ingrid Eshun-Wilson, Elvin Geng.

**Project administration:** Michelle Moghadassi, Cirilus Ogollah, Jayne Lewis Kulzer, Julie Kadima, Francesca Odhiambo, Craig R. Cohen, Elvin Geng.

**Resources:** Jayne Lewis Kulzer, Francesca Odhiambo, Craig R. Cohen, Elvin Geng.

**Software:** Michelle Moghadassi, Elvin Geng.

**Supervision:** Eric Juma, Cirilus Ogollah, Francesca Odhiambo, Elizabeth A. Bukusi, Elvin Geng.

**Validation:** Laura Packel, Ingrid Eshun-Wilson, Hae-Young Kim, Elizabeth A. Bukusi.

**Visualization:** Elvin Geng.

**Writing – original draft:** Raphael Onyango Mando.

**Writing – review & editing:** Raphael Onyango Mando, Laura Packel, Jayne Lewis Kulzer, Francesca Odhiambo, Ingrid Eshun-Wilson, Hae-Young Kim, Elvin Geng.

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
