## [Decision Letter · Decision Letter 0]

13 Jul 2022

PGPH-D-21-01122

Patient preferences for HIV service delivery models; A Discrete Choice Experiment in Kisumu, Kenya

Dear Dr. Onyango,

Thank you for submitting your manuscript to PLOS Global Public Health. After careful consideration, we feel that it has merit but does not fully meet PLOS Global Public Health’s publication criteria as it currently stands. Therefore, we invite you to submit a revised version of the manuscript that addresses the points raised during the review process.

Overall this paper is well-written and represents an important contribution to the growing literature on evidence for HIV treatment delivery models, and this is a unique contribution to incorporating local patient preferences to inform the structure of HIV delivery options in Kenya. The strengths of the manuscript are its topical importance, the use of innovative methods to assess patient preferences, and the thoughtful application of findings to considerations for program design.

The main weaknesses of the manuscript are that it lacks references for the background, lacks sufficient description in some places of interpretation of findings from methods that will be novel to most readers, and in other places spends too much time on details that may be best allocated to a supplemental appendix or tables. As a result, the paper becomes repetitious and unnecessarily wordy at times. Minor grammatical errors throughout--suggest a final read-over for grammar & punctuation. All concerns are minor and addressable. Below are enumerated comments to address:

Major comments:

1. Lines 81-110, Background: Important background to set up the motivation for the study; please provide references. No references are cited in the entire background section. (Note -- it looks like you have these references in your reference list, they just never appear enumerated in the backround).

2. Line 130: Consider using PLHIV instead of "HIV-positive clients." You go on to use person-first language to describe people with HIV; recommend starting with it here.

3. Line 134: What models of DSD are available to clients at these centers? ie how much of a ppt's own experience might inform their choices in the DCE, or were most of the options seen as largely theoretical/ideas that they had not been exposed to in practice?

4. Line 583, Table 2: Suggest limiting to 2-3 significant figures for percents & CIs. Also, in the % column, you use decimals but I believe you want to use whole percents. (e.g. Income Below 5000 (unit? Ks/month?) N=129, %=54.4 (not 0.5443).

Overall for description of the analytic methods, it might be helpful to make the descriptions more concise. Use references to the methods you use, rather than describing precisely how each method works. If you wish to describe methods more extensively, consider placing a longer description of the methods, model-fitting, and selection process into a supplemental appendix.

5. Lines 217 - 234 Willingness to Trade Analysis: This section is very wordy and sometimes repetitive. For example you use the phrase "we fit our final model..." twice. Can you streamline the writing of this paragraph?

6. Line 131 - is there a reference for the non-traditional WTA?

7. Results: In the text of the results, it is helpful to place the reference to the relevant figure or table as close as possible to the beginning of the paragraph that discusses them (rather than at the end of the paragraph).

8. Lines 267 - 278. Can you give some context for the magnitude of the preference weights? You note that the strongest preference was provider attitude; is a beta of 1.61 considered large? Given the confidence intervals overlap between the B for provider attitude and the B for ART refill location, are these in fact significantly different, or is the relative strength of the preference for these two attributes of similar importance? Interpreting this for the reader is probably more important than providing a lengthy explanation of how the model was created.

9. Line 280 - Not clear to the reader how the coeffecients presented reflect the willingness to trade? Consider changing the methods to describe what the coefficients mean (how to interpret them) rather than describing at length the models that generate them.

10. Same for LCA results -- what is the significance of the delta-Betas?

11. Discussion: start by framing the discussion with a summary statement of what you did -- "In this study of PLHIV patient preferences for DSD components, respondents had the strongest preference...."

12. Line 320. What did the study in Ghana show? Helpful to give a 1-2 sentence summary to demonstrate the consistence or discordance of the results.

Also, do you find this surprising? In many places where CAGs are used, CAG participants are quite happy to receive refills in the CAG setting outside of the facility. Did participants who were respondents in the DCE have exposure to CAGs? Could it be that the reason that facilities were the preferred refill setting was because there was a lack of familiarity with the other options?

Line

Minor concerns:

1. Line 96: Editing error "[describe an example of two very briefly here]" should be deleted and replaced with description.

2. Line 98: delete "for example;" unnecessarily wordy.

Line 100: "Instead, the literature needs research..." The literature doesn't need research; national programs, implementers, and funders need research.

3. Line 102: 1st sentence - very wordy. Can you shorten to make more concise?

4. Line 110: please define abbreviations DSD, LMIC in the first instance

5. Line 113: Please define abbreviation DCE in the first instance in the text, then can use abbreviation after

6. Line 158-161. Not clear how multi-month scripting (described as "another component of the care model") is different from reduced frequency of ART pick-ups. Are these in fact different?

7. Line 168 - consider giving more examples of types of people providing ART refills -- clinician, pharmacy, peer, or self-collect e.g. through vending machine.

8. Line 182. Instead of providing Table 1 here in the first instance, suggest providing the Table 1 link at the beginning of this section, so the reader can follow along with the Table as they read more about each attribute/level.

9. Line 198 - avoid using emdashes in writing (e.g. put "expressed as a percentage" in regular text, at the end of the sentence (or in parentheses), or consider eliminating entirely. Same for Line 202 "generated from the attributes."

10. Line 202 - is there a reference for this method that you can cite?

11. Line 253 "The median age among the respondents was 38 (31-44)." Is 31-44 an IQR? Range? Please specify in text.

12. Results -- again, limit %s to one digit after decimal place.

13. Line 260 - repetitive to say that "about 20%" of respondents selected location of ART refills...and then provide the exact percentage. Just use one percentage (ideally the exact number).

14. Line 662 - Table 5 - p-values should not be 0.000 anywhere -- if that is the software output, please present as <0.001 (p-values are never truly 0!)

15. Line 665 - Are these Incidence Rate Ratios as the table header suggests? or RRR, as the column header suggests?

16. Line 323 - Referencing

17. Line 325 "resource-limited settings" instead of "poor resource settings"

18. Line 354 - Agree with your strengths, but also important to include some limitations. One potential limitation you should address is how generalizable these results are. What do you think is the relevant application for your results for DSD program design? Kenya? Outside of Kenya? You give many examples in the discussion of DSD models in diverse locations such as Ghana, Zimbabwe, and China. Do you think the settings are comparable? As you make your conclusions, important to reflect on the generalizability of your findings. (It's not a flaw of the study if the results only apply to Kenya, or to one region of Kenya, but it should be acknowledged).

Reviewer 2

The methodology appears quite rigorous but may need some improvement particularly the study design which lacks detail. Explain in more detail how the Discrete Choice Experiment(DCE) was conducted . Also What's your exclusion criteria?

Reviewers' comments:

Reviewer's Responses to Questions

**Comments to the Author**

1. Does this manuscript meet PLOS Global Public Health’s publication criteria? Is the manuscript technically sound, and do the data support the conclusions? The manuscript must describe methodologically and ethically rigorous research with conclusions that are appropriately drawn based on the data presented.

Reviewer #1: Yes

Reviewer #2: Partly

2. Has the statistical analysis been performed appropriately and rigorously?

Reviewer #1: I don't know

Reviewer #2: Yes

3. Have the authors made all data underlying the findings in their manuscript fully available (please refer to the Data Availability Statement at the start of the manuscript PDF file)?

Reviewer #1: No

Reviewer #2: Yes

4. Is the manuscript presented in an intelligible fashion and written in standard English?

Reviewer #1: Yes

Reviewer #2: Yes

5. Review Comments to the Author

Reviewer #1: To the authors:

Thank you for the opportunity to review this manuscript on patient preferences for HIV health care delivery, assessed by a discrete choice experiment. Overall this paper is well-written and represents an important contribution to the growing literature on evidence for HIV treatment delivery models, and this is a unique contribution to incorporating local patient preferences to inform the structure of HIV delivery options in Kenya. The strengths of the manuscript are its topical importance, the use of innovative methods to assess patient preferences, and the thoughtful application of findings to considerations for program design.

The main weaknesses of the manuscript are that it lacks references for the background, lacks sufficient description in some places of interpretation of findings from methods that will be novel to most readers, and in other places spends too much time on details that may be best allocated to a supplemental appendix or tables. As a result, the paper becomes repetitious and unnecessarily wordy at times. Minor grammatical errors throughout--suggest a final read-over for grammar & punctuation. All concerns are minor and addressable. Below are enumerated comments to address:

Major comments:

1. Lines 81-110, Background: Important background to set up the motivation for the study; please provide references. No references are cited in the entire background section. (Note -- it looks like you have these references in your reference list, they just never appear enumerated in the backround).

2. Line 130: Consider using PLHIV instead of "HIV-positive clients." You go on to use person-first language to describe people with HIV; recommend starting with it here.

3. Line 134: What models of DSD are available to clients at these centers? ie how much of a ppt's own experience might inform their choices in the DCE, or were most of the options seen as largely theoretical/ideas that they had not been exposed to in practice?

4. Line 583, Table 2: Suggest limiting to 2-3 significant figures for percents & CIs. Also, in the % column, you use decimals but I believe you want to use whole percents. (e.g. Income Below 5000 (unit? Ks/month?) N=129, %=54.4 (not 0.5443).

Overall for description of the analytic methods, it might be helpful to make the descriptions more concise. Use references to the methods you use, rather than describing precisely how each method works. If you wish to describe methods more extensively, consider placing a longer description of the methods, model-fitting, and selection process into a supplemental appendix.

5. Lines 217 - 234 Willingness to Trade Analysis: This section is very wordy and sometimes repetitive. For example you use the phrase "we fit our final model..." twice. Can you streamline the writing of this paragraph?

6. Line 131 - is there a reference for the non-traditional WTA?

7. Results: In the text of the results, it is helpful to place the reference to the relevant figure or table as close as possible to the beginning of the paragraph that discusses them (rather than at the end of the paragraph).

8. Lines 267 - 278. Can you give some context for the magnitude of the preference weights? You note that the strongest preference was provider attitude; is a beta of 1.61 considered large? Given the confidence intervals overlap between the B for provider attitude and the B for ART refill location, are these in fact significantly different, or is the relative strength of the preference for these two attributes of similar importance? Interpreting this for the reader is probably more important than providing a lengthy explanation of how the model was created.

9. Line 280 - Not clear to the reader how the coeffecients presented reflect the willingness to trade? Consider changing the methods to describe what the coefficients mean (how to interpret them) rather than describing at length the models that generate them.

10. Same for LCA results -- what is the significance of the delta-Betas?

11. Discussion: start by framing the discussion with a summary statement of what you did -- "In this study of PLHIV patient preferences for DSD components, respondents had the strongest preference...."

12. Line 320. What did the study in Ghana show? Helpful to give a 1-2 sentence summary to demonstrate the consistence or discordance of the results.

Also, do you find this surprising? In many places where CAGs are used, CAG participants are quite happy to receive refills in the CAG setting outside of the facility. Did participants who were respondents in the DCE have exposure to CAGs? Could it be that the reason that facilities were the preferred refill setting was because there was a lack of familiarity with the other options?

Line

Minor concerns:

1. Line 96: Editing error "[describe an example of two very briefly here]" should be deleted and replaced with description.

2. Line 98: delete "for example;" unnecessarily wordy.

Line 100: "Instead, the literature needs research..." The literature doesn't need research; national programs, implementers, and funders need research.

3. Line 102: 1st sentence - very wordy. Can you shorten to make more concise?

4. Line 110: please define abbreviations DSD, LMIC in the first instance

5. Line 113: Please define abbreviation DCE in the first instance in the text, then can use abbreviation after

6. Line 158-161. Not clear how multi-month scripting (described as "another component of the care model") is different from reduced frequency of ART pick-ups. Are these in fact different?

7. Line 168 - consider giving more examples of types of people providing ART refills -- clinician, pharmacy, peer, or self-collect e.g. through vending machine.

8. Line 182. Instead of providing Table 1 here in the first instance, suggest providing the Table 1 link at the beginning of this section, so the reader can follow along with the Table as they read more about each attribute/level.

9. Line 198 - avoid using emdashes in writing (e.g. put "expressed as a percentage" in regular text, at the end of the sentence (or in parentheses), or consider eliminating entirely. Same for Line 202 "generated from the attributes."

10. Line 202 - is there a reference for this method that you can cite?

11. Line 253 "The median age among the respondents was 38 (31-44)." Is 31-44 an IQR? Range? Please specify in text.

12. Results -- again, limit %s to one digit after decimal place.

13. Line 260 - repetitive to say that "about 20%" of respondents selected location of ART refills...and then provide the exact percentage. Just use one percentage (ideally the exact number).

14. Line 662 - Table 5 - p-values should not be 0.000 anywhere -- if that is the software output, please present as <0.001 (p-values are never truly 0!)

15. Line 665 - Are these Incidence Rate Ratios as the table header suggests? or RRR, as the column header suggests?

16. Line 323 - Referencing

17. Line 325 "resource-limited settings" instead of "poor resource settings"

18. Line 354 - Agree with your strengths, but also important to include some limitations. One potential limitation you should address is how generalizable these results are. What do you think is the relevant application for your results for DSD program design? Kenya? Outside of Kenya? You give many examples in the discussion of DSD models in diverse locations such as Ghana, Zimbabwe, and China. Do you think the settings are comparable? As you make your conclusions, important to reflect on the generalizability of your findings. (It's not a flaw of the study if the results only apply to Kenya, or to one region of Kenya, but it should be acknowledged).

Reviewer #2: The methodology appears quite rigorous but may need some improvement particularly the study design which lacks detail. Explain in more detail how the Discrete Choice Experiment(DCE) was conducted . Also What's your exclusion criteria?

We look forward to receiving your revised manuscript.

Kind regards,

Tsitsi G. Monera-Penduka

Academic Editor

Journal Requirements:

2.  Please amend your detailed online Financial Disclosure statement. This is published with the article. It must therefore be completed in full sentences and contain the exact wording you wish to be published.

3. Please update your online Competing Interests statement. If you have no competing interests to declare, please state: “The authors have declared that no competing interests exist.”

4. In the online submission form, you indicated that your data will be submitted to a repository upon acceptance. We strongly recommend all authors deposit their data before acceptance, as the process can be lengthy and hold up publication timelines. Please note that, though access restrictions are acceptable now, your entire data will need to be made freely accessible if your manuscript is accepted for publication. This policy applies to all data except where public deposition would breach compliance with the protocol approved by your research ethics board. If you are unable to adhere to our open data policy, please kindly revise your statement to explain your reasoning and we will seek the editor's input on an exemption. Please be assured that, once you have provided your new statement, the assessment of your exemption will not hold up the peer review process.

5. Please provide separate figure files in .tif or .eps format and remove any figures embedded in your manuscript file. Please also ensure that all files are under our size limit of 10MB.

6. We have noticed that you have uploaded Supporting Information files, but you have not included a list of legends. Please add a full list of legends for your Supporting Information files after the references list.

---

## [Decision Letter · Decision Letter 1]

27 Sep 2022

Patient preferences for HIV service delivery models; A Discrete Choice Experiment in Kisumu, Kenya

PGPH-D-21-01122R1

Dear Dr Raphael Mando Onyango,

We are pleased to inform you that your manuscript 'Patient preferences for HIV service delivery models; A Discrete Choice Experiment in Kisumu, Kenya' has been provisionally accepted for publication in PLOS Global Public Health.

Best regards,

Jeannine Uwimana-Nicol, Ph.D.

Academic Editor

Reviewer Comments (if any, and for reference):

Reviewer's Responses to Questions

**Comments to the Author**

1. If the authors have adequately addressed your comments raised in a previous round of review and you feel that this manuscript is now acceptable for publication, you may indicate that here to bypass the “Comments to the Author” section, enter your conflict of interest statement in the “Confidential to Editor” section, and submit your "Accept" recommendation.

Reviewer #1: All comments have been addressed

Reviewer #2: All comments have been addressed

2. Does this manuscript meet PLOS Global Public Health’s publication criteria? Is the manuscript technically sound, and do the data support the conclusions? The manuscript must describe methodologically and ethically rigorous research with conclusions that are appropriately drawn based on the data presented.

Reviewer #1: Yes

Reviewer #2: Yes

3. Has the statistical analysis been performed appropriately and rigorously?

Reviewer #1: I don't know

Reviewer #2: Yes

4. Have the authors made all data underlying the findings in their manuscript fully available (please refer to the Data Availability Statement at the start of the manuscript PDF file)?

Reviewer #1: No

Reviewer #2: Yes

5. Is the manuscript presented in an intelligible fashion and written in standard English?

Reviewer #1: Yes

Reviewer #2: Yes

6. Review Comments to the Author

Reviewer #1: Very nice job with addressing comments. No further concerns. I look forward to seeing the publication.

Reviewer #2: The authors seem to have addressed all the issues raised by me and my co-reviewers.

7. PLOS authors have the option to publish the peer review history of their article (what does this mean?). If published, this will include your full peer review and any attached files.

**Do you want your identity to be public for this peer review?** For information about this choice, including consent withdrawal, please see our Privacy Policy.

Reviewer #1: No

Reviewer #2: **Yes: **Sandra Shawarira-Bote
